# Genetic Diversity and Population Structure Analysis of *Castanopsis hystrix* and Construction of a Core Collection Using Phenotypic Traits and Molecular Markers

**DOI:** 10.3390/genes13122383

**Published:** 2022-12-16

**Authors:** Na Li, Yuanmu Yang, Fang Xu, Xinyu Chen, Ruiyan Wei, Ziyue Li, Wen Pan, Weihua Zhang

**Affiliations:** 1Guangdong Academy of Forestry, Guangdong Key Laboratory of Forest Cultivation, Protection and Utilization, Guangzhou 510520, China; 2College of Horticulture and Landscape Architecture, Zhongkai College of Agriculture and Engineering, Guangzhou 510225, China; 3Guangdong Forest Resource Conservation Center, Guangzhou 510173, China; 4College of Forestry and Landscape Architecture, South China Agricultural University, Guangzhou 510642, China; 5Academy of Forestry, Hebei Agricultural University, Baoding 071000, China

**Keywords:** *Castanopsis hystrix*, genetic diversity, population genetics, core collection, phenotypic traits, molecular markers

## Abstract

*Castanopsis hystrix* is a valuable native, broad-leaved, and fast-growing tree in South China. In this study, 15 phenotypic traits and 32 simple sequence repeat (SSR) markers were used to assess the genetic diversity and population structure of a natural population of *C. hystrix* and to construct a core germplasm collection by a set of 232 accessions. The results showed that the original population of *C. hystrix* had relatively high genetic diversity, with the number of alleles (Na), effective number of alleles (Ne), observed heterozygosity (Ho), expected heterozygosity (He), Shannon’s information index (I), and polymorphism information content (PIC) averaging at 26.188, 11.565, 0.863, 0.897, 2.660, and 0.889, respectively. Three sub-populations were identified based on a STRUCTURE analysis, indicating a strong genetic structure. The results from the phylogenetic and population structures showed a high level of agreement, with 232 germplasms being classified into three main groups. The analysis of molecular variance (AMOVA) test indicated that 96% of the total variance was derived from within populations, which revealed a low differentiation among populations. A core collection composed of 157 germplasms was firstly constructed thereafter, of which the diversity parameters non-significantly differed from the original population. These results revealed the genetic diversity and population structure of *C. hystrix* germplasms, which have implications for germplasm management and genome-wide association studies on *C. hystrix*, as well as for core collection establishment applications in other wood-producing hardwood species.

## 1. Introduction

Information on genetic diversity is important for understanding the extent of genetic variability in existing plant material and the breeding and conservation of genetic resources [1,2]. However, tree breeding usually involves the recurrent selection of genetically superior materials and possibly results in altered diversity levels in breeding populations [3]. Various types of markers can be used for genetic diversity estimation. In the past, phenotypic traits were widely used for assessing genetic diversity; however they are influenced by the environment and cannot be accurately evaluated. In recent decades, DNA molecular markers have been increasingly exploited for genetic diversity. They can be employed to investigate levels of genetic diversity among categories such as cultivars and closely related species in germplasm banks [4,5].

The collection and preservation of germplasm resources are of great significance to genetic improvement, new variety breeding, and germplasm innovation of forest trees [6]. Brown [7,8] and Frankel and Brown [9] first proposed the concept of core collection, an approach using a minimum number of germplasm resources from the whole germplasm bank to represent the maximum genetic diversity of the original collection. The construction of a core collection provided a new approach for the in-depth evaluation, efficient protection, and utilization of germplasm resources [10,11,12] and has gradually become a hot topic in international germplasm resources research [13]. The research on core collection in China started in 1994 and was first applied to some crops [14,15], while applications in trees were relatively rare. Using molecular markers and phenotypic traits to construct a core collection, the full genetic diversity of tree species can be better retained, resulting in improved accuracy and reliability, as shown in *Cryptomeria japonica* [16] and *Robinia pseudoacacia* [17]. Recent theoretical studies on sampling ratio, sampling strategy, and effectiveness evaluation of core collection further advance the field by providing a theoretical basis for the construction and representative evaluation of core collections [18,19,20,21].

*Castanopsis hystrix* is an evergreen broad-leaved tree in the Fagaceae mainly distributed in Guangdong, Guangxi, Fujian and other southern provinces of China. It is widely used in furniture, construction, and shipbuilding due to its fast growth and good adaptability to various materials. *C. hystrix* has been identified as a precious species in the Guangdong province. Since 1999, Guangdong has continuously collected germplasm materials of *C. hystrix* from different habitats. Aimed at developing effective and efficient conservation practices for plant genetic resources, understanding the genetic diversity between and within the population is important [22,23,24]. Assessing relatedness among accessions is an important prerequisite for the identification of core collections suitable for optimizing association studies [25]. The analysis of genetic diversity and population genetic structures is significant for verifying domestication events and genetic relationships of *C. hystrix*. In the past, molecular markers, i.e., random amplified polymorphic DNA (RAPD) [26] and inter-simple sequence repeat (ISSR) [27,28,29] have been applied to assess the genetic diversity of *C. hystrix* resources. Molecular markers provide a powerful tool for genetic diversity analysis and core collection establishment. Of the marker types, SSR markers consist of 2–6 bp nucleotide repeat motifs, which are considered to be one of the most effective molecular markers for studying genetic diversity. Its codominance, abundant polymorphism information, and genomes specify and show a fairly even distribution over the genome [30]. SSR markers have found applications in the analyses of genetic diversity and population structures, gene mapping, and assisted selection for crop improvement [31,32,33,34]. However, SSR markers have been less frequently applied in *C. hystrix*, and there is no relevant research on the construction of a *C. hystrix* core collection.

In this study, to ascertain the genetic diversity and population structure and to construct a core collection, a total of 232 *C. hystrix* accessions from 17 provenances were analyzed using 32 SSR markers distributed throughout the *C. hystrix* genome and using 15 phenotypic traits in the whole distribution area. Our objectives were to estimate the levels of genetic diversity and to characterize the population structure of the *C. hystrix* germplasm collection. The results are intended to provide a molecular basis for understanding *C. hystrix* genetic diversity, effectively preserving and utilizing *C. hystrix* germplasm resources, which provide better materials for *C. hystrix* breeding and ensure the population inheritance of important traits.

## 2. Materials and Methods

### 2.1. Experimental Materials

The germplasm gene bank was established in 2003. Since 1999, the *C. hystrix* Genetic Improvement Research Collaboration Group, composed of the Guangdong Academy of Forestry and other institutions, has systematically investigated, selected, and sampled superior trees from 17 provenance areas with relatively concentrated distributions of *C. hystrix* germplasm resources according to the distribution of existing *C. hystrix* resource. Furthermore, the group has also carried out the collection of *C. hystrix* germplasm resources. Superior trees were selected by the method of five dominant trees, and one tree was selected for every 30–50 m. The number of superior trees selected from each producing area was generally 10–30 trees according to the distribution area. After selecting excellent trees, we collected seeds and the sunny branches of the specific year in the south side of the middle and upper part of the crown from October to November 2001. After sending them directly to the grafting site, 30–50 plants were grafted on each excellent tree. The grafted trees were transplanted in the gene bank in the spring of 2003, and five plants were planted in each excellent tree; the plant row spacing was 3 m × 5 m, and the size of the hole was 60 cm × 60 cm × 40 cm.

A set of 232 accessions of *C. hystrix* were mainly collected from the whole range (Figure 1), which mainly comprised 17 provenances, including Guangxi (5 accessions), Guangdong (4 accessions), Fujian (3 accessions), Hainan (2 accessions), Yunnan (2 accessions), and Hunan (1 accessions). At present, all of them are preserved in the Maofeng Mountain *C. hystrix* germplasm gene bank (113° 46′ E and 23° 29′ N), Baiyun District, Guangzhou City, Guangdong Province. From November to December 2018, 3 trees were selected for each clone, and 15 phenotypic traits were investigated, including 5 growth traits, 2 morphological traits, and 8 wood properties (Appendix A); the detailed passport data is presented below (Table 1). The laboratory had completed the DNA extraction and SSR genotyping in the early stage. Finally, 32 pairs of SSR markers were chosen (Appendix A). For the specific operation methods and steps of the test, refer to the paper by Yang [35].

### 2.2. Genetic Diversity Analysis

The format conversion software Convert v1.31 [36] was used to convert the results into the POPGENE format. Assessments of genetic diversity, including observed heterozygosity (Ho), expected heterozygosity (He), and polymorphic information content (PIC), were estimated using CERVUS v3.0.7 software [37,38]. The number of alleles (Na), number of effective alleles (Ne), Shannon’s information index (I), genetic differentiation index (Fst), and gene flow (Nm) were calculated using GenAlEx v6.5 software [39,40].

### 2.3. Population Structure, Principal Coordinate Analysis, and Evolutionary Tree Analysis

An analysis of molecular variance (AMOVA) test was carried out to determine the relative partitioning of the total genetic variation among and within different groups of genotypes by using GenAlEx 6.5. The principal coordinate analysis (PCoA) was also performed using GenAlEx v6.5. The genetic structure of unique genotypes was investigated using STRUCTURE v2.3.4 software [41] using an admixed model with 10,000 burn-ins followed by 10,000 iterations. Markov Chain Monte Carlo iterations were run for 20 cycles of a number (K = 1–10) of genetically homogeneous clusters. The most probable K value was determined with the highest ΔK method [42] in STRUCTURE HARVESTER v0.6 software [43] and used for the estimation of the membership coefficient of each clone. The web tool iTol (https://itol.embl.de/ (accessed on 2 July 2021)) was used for data visualization. Additionally, to analyze the relationships of the 232 germplasms, a genetic distance matrix between the clones was generated, and an unrooted phylogenetic tree was constructed using the neighbor-joining method in PowerMarker 3.25 software [44].

### 2.4. Construction of the Core Collection and Evaluation

QGAStation v2.0 software(Guobo Chen, Futao Zhang and Jun Zhu, Zhejiang University, China) was used to construct the core collection based on phenotypic data, combining three systematic clustering methods (unweighted pair-group average method, Ward’s method, and median distance method), two genetic distances (Euclidean distance and Mahalanobis distance), three sampling methods (random sampling, deviation sampling, and preferred sampling) and seven sampling ratios (10%, 15%, 20%, 25%, 30%, 35%, and 40%) [45].

Core Finder v1.1 [46] and Core Hunter 3 [47] software were used to establish a core collection according to the molecular markers data; the latter software set various sampling ratios. The independent *t*-test was used to analyze the significance of differences in genetic diversity parameters between the core collection and the original collection [48]. If the differences were not significant, the constructed core collection was considered to be representative of the original collection. The principal coordinates analysis (PCoA) was performed to generate the distribution map of the core and original collection to evaluate the core collection.

## 3. Results

### 3.1. Genetic Diversity Analysis of the Original Population

A total of 335 alleles were detected by 32 polymorphic microsatellite markers, with an average of 10.458 alleles detected at each locus, among which 15 alleles were detected at the SSR04 locus. The alleles of SSR02, SSR10, SSR15, SSR18, SSR23, and SSR27 were the same and least, having only eight alleles (Table 2). The average number of effective alleles (Ne) at all loci was 7.115, ranging from 3.832 to 10.931. The average Shannon diversity index (I) was 2.035, ranging from 1.597 to 2.469. The average observed heterozygosity (Ho) was 0.861, ranging from 0.736 to 0.990. The average expected heterozygosity (He) of the 32 loci was 0.824, ranging from 0.704 to 0.894, indicating that the *C. hystrix* had a high level of diversity. The average polymorphism information index (PIC) was 0.889, ranging from 0.744 to 0.958. According to the standard of PIC ≥ 0.5 [49], all of the above loci showed a high polymorphism. The average coefficient of genetic differentiation (Fst) of all loci in this study was 0.081, ranging from 0.056 to 0.138, indicating slight genetic differentiation in the loci. The average value of gene flow (Nm) was 2.948, ranging from 1.561 to 4.200, indicating that the gene flow greatly fluctuated in different loci, which indicates that there was a high degree of gene communication among the *C. hystrix* populations. The results showed that the genetic diversity of these SSR loci was generally high, and the highest genetic diversity was SSR04 while the lowest genetic diversity was SSR10, observed by combining the values of each genetic parameter.

In order to analyze the genetic diversity of 17 *C. hystrix* populations, the genetic parameters, containing Na, Ne, Ho, He, and I were calculated, respectively (Table 3). The average number of alleles (Na) was 10, ranging from 3 to 15. The average number of effective alleles (Ne) was 7.115, ranging from 2.821 to 9.519. The average Shannon diversity index (I) was 2.035, ranging from 1.025 to 2.395. The mean observed heterozygosity (Ho) was 0.861, ranging from 0.766 to 0.923. The average expected heterozygosity (He) was 0.824, ranging from 0.609 to 0.876. These results indicated that the overall genetic diversity of *C. hystrix* was high and that the level of variation was rich.

### 3.2. Population Structure of the Original Population

The population structure of the 232 accessions was estimated using STRUCTURE software based on the 32 SSR markers. Firstly, the number of subpopulations (K) was identified based on the maximum likelihood and DK values; the results showed that the DK value reached the highest when K = 3, which indicated that the whole population was divided into three subgroups (Figure 2). The three subgroups were designated as Q1, Q2, and Q3 (indicated in red, green, and blue, respectively, in Figure 3). At K = 3, the division was as follows: group I included 100 accessions; group II contained 57 accessions; and group III contained 75 accessions (Appendix A). There was an admixture that occurred between the clusters, indicating that there was a certain degree of gene exchange among the populations.

A principal component analysis was performed to create a three-dimensional scatter plot using the data of the SSRs identified in the 232 *C. hystrix* germplasms to visualize the relationships between genotypes. A three-dimensional graph was created based on the value of each sample in the first (PC1), second (PC2), and third (PC3) principal components (Figure 4). The first, second, and third principal components explained 8.4%, 6.3%, and 4.8% of the total genetic variability, respectively. The scattered dots of different colors in the PCA figure represent samples of different populations, and the results show that the accessions are clustered together, indicating that the differences of these accessions are small.

A neighbor-joining analysis was performed, and the 232 germplasms were classified into three main groups, designated as groups I, II, and III (Figure 5). The distribution of the *C. hystrix* germplasms in the inferred groups is shown in Appendix A. The first main group (I) included 61 germplasms. The second main group (II) included 68 germplasms. The third main group (III) included 103 germplasms. From a geographic origin perspective, some of the germplasms from the same geographic origin clustered in the different group (Appendix A). This shows that the three groups classified by phylogenetic analysis contained germplasms from different geographical locations. The neighbor-joining dendrogram based on the genetic distance between individual trees was used to determine the genetic relationship among *C. hystrix* accessions, and a similar result of structure analysis at K = 3 was obtained.

In order to understand the level of genetic differentiation and reflect the source of variation among *C. hystrix* populations, the source of variation was divided into two levels (between different populations and within populations), and an AMOVA analysis of molecular variance (Table 4) was performed on the *C. hystrix* populations. The results showed that the genetic variation of *C. hystrix* populations mainly came from individuals within the population, and most of the variation was within populations (96%), whereas 4% of the variation was between populations.

### 3.3. Core Collection Establishment and Evaluation

Using QGAStation software to construct core collection based on phenotypic data, a total of 126 core collections were constructed. Among the 126 core collections, three with CR < 80% were removed (Figure 6, Appendix A); three core collections had a CR of 10%. Eventually, 123 core collections had a mean difference (MD) of <20% and coincidence rate of range (CR) > 80%, indicating that these 123 core collections are good representations of the genetic diversity of the original collection. According to the maximum variance difference (VD) and rate of variation in the coefficient of variation (VR) values at each sampling ratio (Table 5), we found that the VR had maximum values at the 10% sampling ratio, and the CR and VD had maximum values at the 15% sampling ratio. With the increase in sampling proportion, the VR gradually decreased. Therefore, 15% is the optimal sampling ratio. Under the preferred sampling method (D3), the CR of the core collection constructed was 100%, but the CR constructed by the deviation sampling method (D2) was lower than 100%. Therefore, the best sampling method is the preferred sampling method (D3). At the 15% sampling ratio, the VD value at C3 was greater than C2, and the VR value was similar; thus, the best clustering method was determined to be the mediate distance method (C3). We conclude that the core collection generated from the 15% sampling ratio (B2C3D3) is the best core collection, which has 32 clones.

Core Finder software was used to analyze the SSR data with the M strategy based on the principle of maximizing alleles. The core collection Mc1 having a sample size of 158 retained 100% of the allele number of the original collection and had increased genetic parameters (Ne, He, I, and PIC); thus, Mc1 with its strong genetic diversity should be a good representation of the original collection. In addition, Core Hunter 3 was used to determine the optimal size of the core collection using 10 preset sampling ratios (Table 6). The six genetic diversity parameters (Na, Ne, Ho, He, I, and PIC) of the core collection and the original collection were compared. We found that the core collections with a sampling ratio of less than 50% had significantly different Na and Ho values than the original collection; thus, the optimal sampling ratios is 55% (H-55, referred to henceforth as Mc2). To further determine the better core collection between Mc1 and Mc2, we analyzed the Ne, I, Ho, He, and PIC of the constructed core collection (Table 7). The core collection Mc1 and Mc2 preserved 68.1% and 55.2% of the original collection resources, respectively. The *t*-test results showed that the six genetic diversity parameters of core collection Mc1 and Mc2 were not significantly different from those of the original collection, indicating that both core collections could be a good representation of the original population. Based on the values summarized in both Table 6 and Table 7, the Ne, He, I, and PIC values of Mc1 obtained by Core Finder were all higher than those of the original collection. This indicates that the genetic redundancy in the original collection was removed from Mc1 and the corresponding genetic diversity parameters were increased. Only the He and PIC values of Mc2 obtained by Core Hunter 3 were higher than those of the original collection, and the reservation rates of all six genetic parameters were lower than those of Mc1. Moreover, the proportion of Na retention of Mc1 generated by Core Finder was 99.9%, and the retention rates of Ne, He, I, and PIC were all above 100%; PIC ≥ 0.5 had high polymorphism, indicating it is a good core collection. Therefore, the best core collection was identified as Mc1, which has 158 clones.

Finally, a total of 157 clones were obtained by combining the phenotypic core collection (B2C3D3-15) and molecular core collection (Mc1) into the final core set (BM). In order to check if the BM could effectively represent the genetic diversity of the whole germplasm, a principal coordinates analysis (PCoA) was used to generate a distribution map of the core and original collection with SSR data (Figure 7). The results showed that the distribution of the core collection and original collection basically coincides in the middle part, indicating that this part of the core collection is a good representation of the original collection. However, there was deviation in the upper right and lower right part.

## 4. Discussion

### 4.1. Genetic Diversity of C. hystrix Germplasm Resources

Characterizing breeding collection germplasms is crucial in plant breeding as the genetic advancement of economically valuable traits relies on the genetic diversity available within the breeding gene pool. Learning about genetic diversity also assists in minimizing the use of closely related clones as parents in breeding programs. Genetic diversity is an integral part of all biological diversity; it is the basis of biological evolution and species differentiation and is of great significance for population maintenance, reproduction, and adaptation to habitat changes. The higher the genetic diversity, the more likely a population is to adapt to different environments, and variations in DNA sequences are the primary drivers of such diversity [50]. Molecular markers provide powerful tools for genetic diversity analyses and the establishment of core collections. In recent years, various molecular markers such as RAPD, SSR, and ISSR have been used to study the analysis of phylogeny, inter-species relationships, and genetic diversity of forest species including *Pinus leucodermis*, *Eucalyptus globulus*, *Swietenia macrophylla*, and *Populus deltoides* [51,52,53,54]. It has been reported that SSRs are abundant and ubiquitous in prokaryotic and eukaryotic genomes [55,56]. SSRs offer high-resolution markers to breeding programs far beyond the traditionally used approaches solely depending on pedigrees [7] or phenotypic data [57]. Consequently, SSRs have become the most popular marker.

*C. hystrix* is a precious local wood and an efficient multi-purpose fast-growing tree species in South China. Genetic diversity, population structure, and molecular markers knowledge may accelerate the selection of desirable traits in *C. hystrix*. Nei’s gene diversity, the observed heterozygosity and expected heterozygosity, the Shannon–Wiener index, the polymorphism information content, etc., have all been used to evaluate the level of genetic diversity of plant species [50]. The high number of alleles obtained in some studies may be due to the use of a large amount of highly diversified plant material [58,59] as well as the high number of samples employed in the analysis. In the present study, a total of 335 alleles was revealed using 32 SSR markers, with an average 10 alleles per locus, revealing a high level of variability within a sample set. This high average of alleles per locus can be attributed to the high genetic diversity in the investigated genotypes. The PIC value affords a fairer estimation of diversity than the actual number of alleles because it takes into account the relative frequencies of each allele present [60,61]. In our study, the overall average PIC for the SSR loci value was 0.889. All SSRs had PIC values ranging from 0.744 to 0.958. The SSRs having PIC values ranging from 0.25 to 0.5 are considered moderately informative [49]. This result was also reported for *Xanthoceras sorbifoliai*, and higher PIC and genetic diversity scores were reported in studies using SSRs [62]. According to the genetic diversity of the 17 *C. hystrix* populations, the P2 population (Bobai, Guangxi) had the highest genetic diversity, while the P13 population (Jianghua, Hunan) had the lowest genetic diversity. The genetic diversity of the P13 population was significantly lower than that of the other populations, which may be due to the small number of samples. As previously demonstrated, the SSR assay approach is appropriate for genetic relationship studies [63,64], and it proved to be an efficient tool for the assessment of the genetic diversity of *C. hystrix* and identification of its populations in China.

### 4.2. SSR-Based Genetic Relationships among C. hystrix Germplasm Resources

Population structure is an important component in association mapping analyses between molecular markers and traits. Differences in population genetic structures reflect genetic diversity and convey the adaptation potential of a species to its changing environment [65]. To understand the genetic relationships and population genetic structure of the *C. hystrix* germplasm at the genomic level, the SSR data of 232 germplasms, STRUCTURE analysis, UPGMA cluster analysis, and PCoA analyses were used to thoroughly investigate the genetic structure of *C. hystrix*. Based on the SSRs and multiple analyses, including population structure and phylogenetic analyses, it was confirmed that the within population was clearly clustered to three groups, which is more than previous studies using SSR and ISSR molecular marker analysis [27,66]. Both NJ and Bayesian model-based clustering studies failed to indicate any definitive clustering among the germplasm accessions. Although they were clustered into three groups, the results were somewhat different. This may be caused by different clustering methods. We found that the results of the phylogenetic analysis and genetic population analysis were basically consistent and complemented one another, but they are not completely clustered according to geographical origin. This is mainly because the elite germplasms used in this test were all selected and obtained from local gene pool. Moreover, in the long-term selection process, germplasms from different provinces were introduced or exchanged. Wang et al. [67] reported similar results in a study conducted on 119 *Xanthoceras sorbifolium* accessions.

The AMOVA analysis is a satisfactory grouping criterion for evaluating the variation within and among populations. Most scholars generally believe that the level of genetic diversity in woody plants with wide distribution, perennial, outcrossing, wind-borne seeds, or feeding by birds and animals is higher and that the genetic diversity within the populations is richer than that between populations [68,69,70]. In accordance with the genetic variation between and within the populations was significant (*p* < 0.001), the results indicated a greater within population variation (96%) than between populations (4%), and the genetic variation within populations was the primary source of the total variation. This indicates that there is little genetic differentiation among the populations, which matches the results recorded in previous studies [27,28]. The AMOVA results revealed that the population differentiation between the main genetic variation accounted for the largest proportion of genetic variability (Table 4), which was similar to that of previous studies by Li et al. [66] and Belaj et al. [71,72]. In conclusion, the genetic variation mainly came from the within population variation, the genetic diversity within the population was rich, and the genetic differentiation among the populations was small. This may be so for the following reasons: (1) *C. hystrix* is an outcrossing plant mainly pollinated by wind, its wide distribution provides opportunities for gene recombination and produces rich genetic diversity. At the same time, the gene exchange between populations was promoted and the differentiation between populations was reduced. (2) Plant genetic diversity is related to environmental adaptation [73]. The *C. hystrix* has a wide distribution range and strong adaptability. Under the action of long-term natural selection, a wide range of genetic variation has been produced, resulting in the formation of geographical provenances with different phenotypes and different requirements for environmental conditions. (3) The level of genetic diversity within populations was also influenced by the number of samples [74,75], and the genetic diversity within a population is proportional to the number of samples [76].

### 4.3. Core Collection Construction and Evaluation

In this study, the genetic diversity, population structure, population differentiation and, core collection of *C. hystrix* resources have been evaluated using SSR molecular markers, which identified rich genetic diversity among the *C. hystrix* germplasm within populations. Although the core collections of many trees have been established, the construction of a core collection of *C. hystrix* had not been conducted. According to the method of Hu [20], when MD% ≤ 20% and CR% > 80%, the core collection can be considered to recapitulate the genetic diversity of the original collection. The smaller the MD%, the larger the VD%, CR%, and VR%, and the more representative the core collection. The retention ratio of alleles should be greater than 70%, and the larger the other genetic parameters, the better [77,78]. We successfully established a core collection of *C. hystrix* with a 100% allelic representation based on 15 phenotypic traits and 32 SSR markers.

The rich diversity of different germplasm resources is detailed, the sampling ratio of core collection bank varied, and the sampling ratio of woody plants ranged from 10.00% to 45.00% [13,79,80]. In this study, seven sampling ratios were investigated (10%, 15%, 20%, 25%, 30%, 35%, and 40%), and the final sampling ratio of the best *C. hystrix* phenotype core collection was 15%, which is consistent with previous studies [81]. While phenotypic data can often reflect the genetic diversity of materials, perennial trees are vulnerable to environmental impacts. However, molecular marker technology has the advantages of low cost and fast data acquisition, and it is not affected by external factors. The core collection constructed by combining the genetic diversity and phenotypic variation of the original population [82,83,84] can improve the effectiveness of the constructed core collection.

Based on the M strategy, Core Finder software selects core collections by maximizing the number of alleles at each locus, which can eliminate genetic duplication in materials during construction and screen materials with a large number of alleles and low redundancy. Core Hunter 3 mainly screens the core collection based on maximizing genetic diversity and allele richness, and different sampling ratios can be set. In this study, a *C. hystrix* core collection was constructed using Core Finder and Core Hunter 3 software. The results showed that the retention rates of Core Finder in the four genetic parameters Na, Ne, Ho, and I were higher than those of Core Hunter 3 (Table 7). Moreover, the Ne, He, I, and PIC values of Mc1 obtained by Core Finder were higher than those of the original collection; this was expected as the diversity increases with the elimination of genetically similar accessions during core collection development [85]. In conclusion, Core Finder software is more suitable for the construction of the *C. hystrix* core collection. Consistent with this study, Gong et al. [86] constructed an astragalus core collection based on 380 astragalus samples using different methods, such as the M strategy-based method in Core Finder and stepwise sampling-based method in Core Hunter 3; the authors concluded that Core Finder software combined with the M strategy was the most suitable method for constructing the astragalus core collection. In this study, a core *C. hystrix* germplasm set, BM, was constructed based on 15 entries of phenotypic data and 32 SSR markers, which were composed of 157 *C. hystrix* accessions. The results of the principal component analysis showed that some of the core collection overlapped with the original collection and some were scattered around. The reason may be that when using Core Finder software to analyze SSR data and extract the core collection, the amount of data was too large. In the future, we can try to use other software to construct a core collection based on the SSR data and compare the results.

## 5. Conclusions

In this study, the genetic diversity, population structure, population differentiation, and core collection of *C. hystrix* resources have been evaluated using SSR molecular markers. The results showed that the genetic diversity of these SSR loci was rich. Moreover, *C. hystrix* samples were grouped into three clusters. We successfully established a core collection, BM, by combining 15 phenotypic data and 32 SSR molecular markers. We demonstrated that SSR markers were successful and effective for the assessment of the genetic diversity and structure of the *C. hystrix* populations. The established core collection can be used for future genome association analysis and breeding program research. This study provided a theoretical basis for germplasm resource management as well as the conservation and utilization of *C. hystrix* germplasm resources.

## Figures and Tables

**Figure 1 genes-13-02383-f001:**
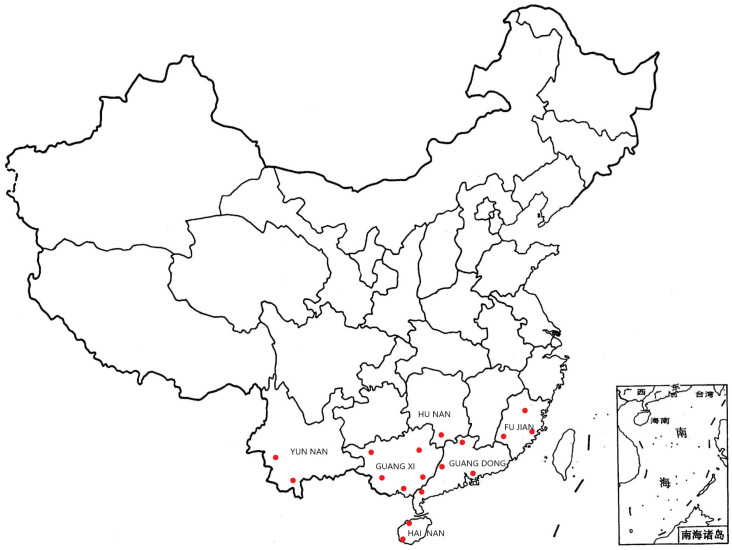
Maps showing the geographic location of the collection site of the 17 *C. hystrix* populations sampled. The red dot in the figure represents 17 sampling sites The Chinese letters in the picture represent six provinces. The lower right corner represents the South China Sea Islands, which are an integral part of China.

**Figure 2 genes-13-02383-f002:**
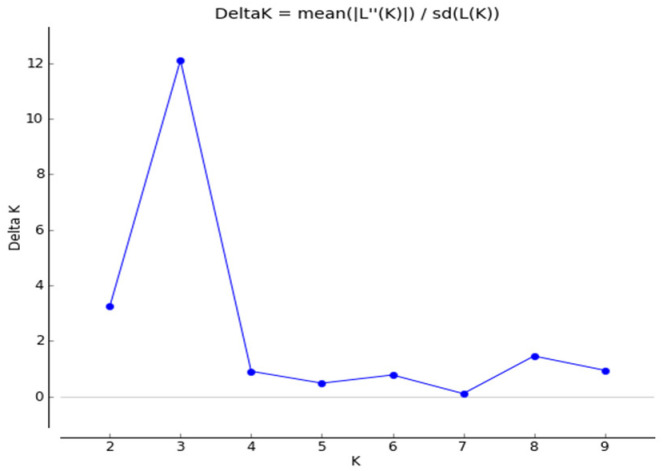
The STRUCTURE estimation of the number of populations for K values ranging from 1 to 10. The K value with the highest delta K represents the suggested cluster number.

**Figure 3 genes-13-02383-f003:**
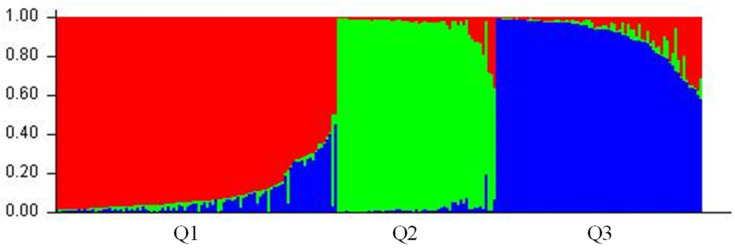
Phylogenetic tree of the accession with three different colors indicating three groups obtained from the STRUCTURE analysis result. Red, green and blue represent Q1, Q2 and Q3 respectively. The vertical coordinate of each subgroup indicates the membership coefficients for each *C. hystrix* accession.

**Figure 4 genes-13-02383-f004:**
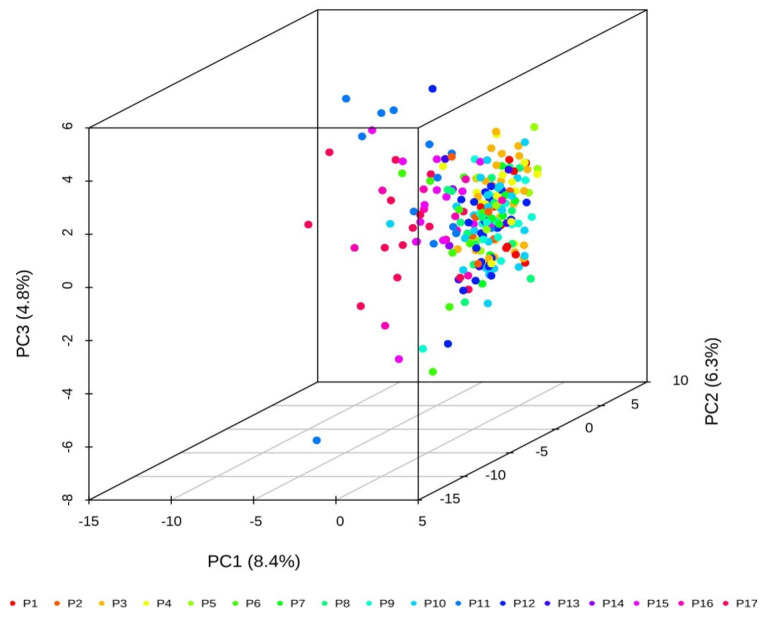
Principal component analysis (PCA) of 232 *C. hystrix* elite germplasms by https://www.bioladder.cn/web/#/chart/62. PCA plot of the first three components (PC1, PC2, and PC3) of the 232 germplasms.

**Figure 5 genes-13-02383-f005:**
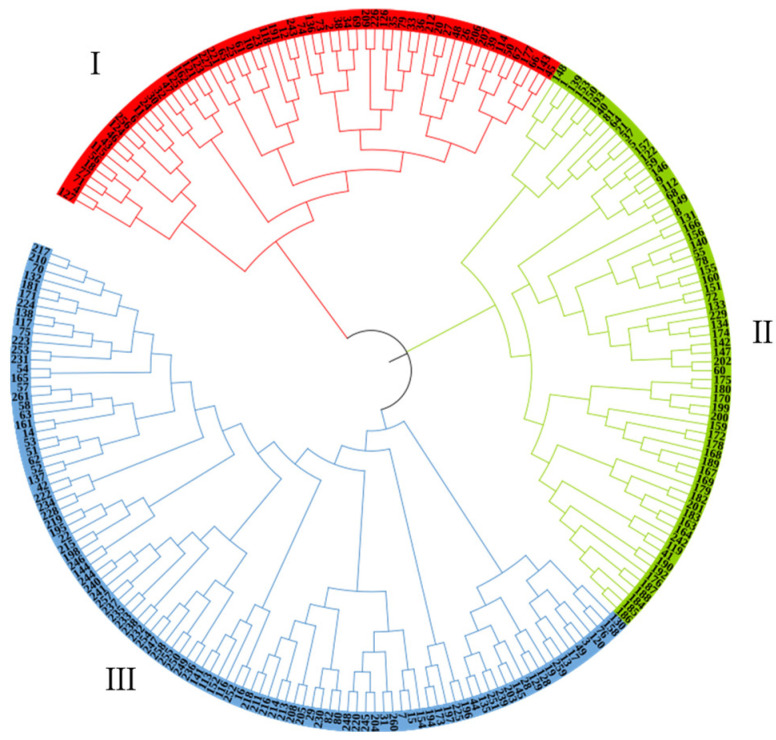
Phylogenetic tree of all 232 *C. hystrix* elite germplasms based on the SSRs built by the neighbor-joining method in PowerMarker 3.25 software. I, II and III represent the three categories of clustering, respectively.

**Figure 6 genes-13-02383-f006:**
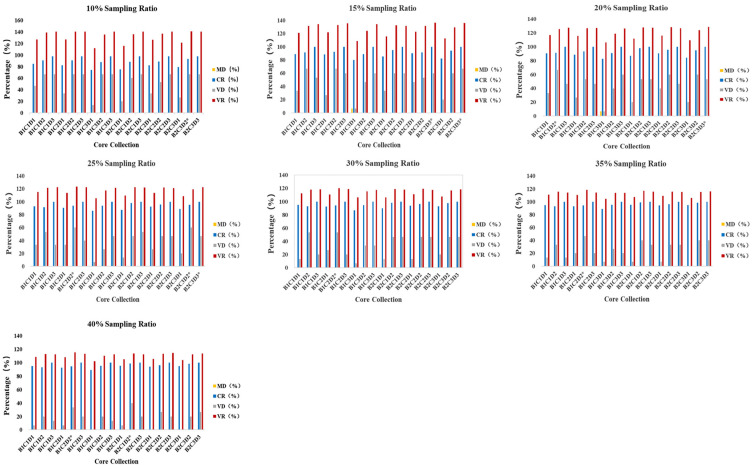
Percentage of trait differences between the core collections and the initial collection obtained by different combinations. B1 and B2 represent Euclidean distance and Mahalanobis distance, respectively; C1, C2, and C3 represent the unweighted pair-group average method, Ward’s method, and mediate distance method in the systematic clustering, respectively. D1, D2, and D3 represent the random sampling, deviation sampling, and preferred sampling methods, respectively. Seven sampling ratios of 10%, 15%, 20%, 25%, 30%, 35%, and 40% were set. The percentage of the mean difference (MD), percentage of the variance difference (VD), coincidence rate of the range (CR) and rate of variation in the coefficient of variation (VR). * represents the maximum for each sampling ratio in the figure.

**Figure 7 genes-13-02383-f007:**
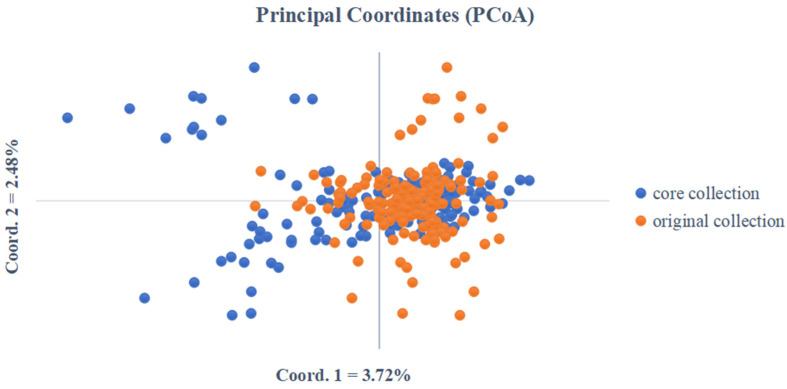
Principal coordinate distribution of the *C. hystrix* core collection and original collection.

**Table 1 genes-13-02383-t001:** Details of sampling points in the 17 province areas of *C. hystrix*.

Site	Group Number	Number of Clones	Longitude and Latitude	Clone Number
Guangxi PuBei	P1	12	109.55° E 22.27° N	A1~A7, F1~F2
Guangxi BoBai	P2	26	109.98° E 22.27° N	B1-1~B10, C8~C9, E12~E16
Guangxi RongXian	P3	15	110.53° E 22.87° N	C1~C7-1, C33, J28~J32
Guangxi PingXiang	P4	9	106.75° E 22.11° N	D1~D8, I39
Guangxi DongLan	P5	9	107.36° E 24.53° N	E1~E10
Guangdong XinYi	P6	17	110.90° E 22.36° N	G1~G14
Guangdong GaoZhou	P7	18	110.50° E 21.54° N	H1~H20
Guangdong LuHe	P8	24	115.65° E 23.30° N	I1~I10, I20~I35, 30, 55
Guangdong ShiXing	P9	8	114.08° E 24.78° N	I11~I18
Fujian JinShan	P10	23	117.37° E 24.52° N	J1~J19
Fujian HuaFeng	P11	5	117.53° E 25.02° N	J20~J21-1
Fujian GaoChe	P12	9	117.39° E 24.31° N	J22~J27
Hunan JiangHua	P13	2	111.79° E 24.97° N	K1~K2
Hainan LeDong	P14	15	109.17° E 18.73° N	L1~L18
Hainan ChangJiang	P15	12	109.03° E 19.25° N	L20~L33
Yunnan JingHong	P16	15	100.79° E 22.00° N	M1~M15
Yunnan SiMao	P17	13	101.00° E 22.79° N	M16~M27, N1
Total		232		232

**Table 2 genes-13-02383-t002:** Genetic diversity parameters of the 32 SSR loci.

Locus	Na	Ne	Ho	He	PIC	I	Fst	Nm
SSR01	11	7.901	0.935	0.862	0.910	2.165	0.056	4.200
SSR02	8	5.894	0.945	0.811	0.866	1.867	0.074	3.119
SSR03	10	7.128	0.856	0.839	0.915	2.045	0.086	2.661
SSR04	15	10.931	0.915	0.894	0.958	2.469	0.068	3.438
SSR05	10	6.565	0.882	0.828	0.881	1.984	0.068	3.453
SSR06	10	6.928	0.741	0.832	0.903	2.028	0.085	2.684
SSR07	10	6.495	0.835	0.822	0.888	2.008	0.073	3.169
SSR08	10	7.128	0.833	0.812	0.895	2.018	0.112	1.990
SSR09	12	9.013	0.955	0.867	0.936	2.275	0.074	3.119
SSR10	8	3.832	0.780	0.704	0.744	1.597	0.094	2.408
SSR11	9	5.202	0.736	0.733	0.833	1.756	0.138	1.561
SSR12	12	8.983	0.966	0.858	0.940	2.248	0.087	2.621
SSR13	10	6.631	0.955	0.824	0.890	2.001	0.085	2.681
SSR14	12	9.039	0.917	0.858	0.937	2.227	0.085	2.674
SSR15	8	4.148	0.775	0.735	0.769	1.620	0.070	3.325
SSR16	10	7.294	0.805	0.840	0.919	2.066	0.089	2.551
SSR17	10	5.742	0.873	0.808	0.851	1.924	0.060	3.924
SSR18	8	5.380	0.809	0.803	0.842	1.831	0.068	3.420
SSR19	10	6.745	0.851	0.825	0.885	2.012	0.079	2.924
SSR20	13	9.786	0.850	0.882	0.948	2.350	0.073	3.191
SSR21	9	6.158	0.909	0.822	0.877	1.919	0.076	3.051
SSR22	12	8.673	0.817	0.854	0.932	2.218	0.090	2.533
SSR23	8	4.943	0.793	0.765	0.830	1.740	0.095	2.387
SSR24	11	7.346	0.942	0.849	0.916	2.100	0.076	3.055
SSR25	12	8.840	0.827	0.863	0.941	2.239	0.083	2.780
SSR26	10	5.715	0.893	0.801	0.859	1.922	0.074	3.121
SSR27	8	4.519	0.753	0.748	0.805	1.664	0.088	2.591
SSR28	13	8.364	0.795	0.841	0.920	2.201	0.079	2.925
SSR29	11	7.897	0.990	0.849	0.904	2.145	0.066	3.538
SSR30	10	6.848	0.902	0.835	0.889	2.032	0.064	3.632
SSR31	11	7.722	0.866	0.841	0.918	2.113	0.086	2.656
SSR32	14	9.880	0.847	0.873	0.943	2.345	0.075	3.079
Mean	10	7.115	0.861	0.824	0.889	2.035	0.081	2.948

**Table 3 genes-13-02383-t003:** Genetic diversity parameters of the 17 populations.

Population	Na	Ne	Ho	He	I
Pop1	11	8.067	0.904	0.858	2.200
Pop2	15	9.519	0.896	0.876	2.395
Pop3	12	8.473	0.877	0.868	2.267
Pop4	9	6.835	0.923	0.835	2.023
Pop5	9	6.516	0.902	0.836	1.986
Pop6	12	8.097	0.859	0.856	2.217
Pop7	13	8.910	0.885	0.866	2.306
Pop8	14	8.669	0.853	0.871	2.324
Pop9	8	5.972	0.887	0.815	1.886
Pop10	13	7.639	0.866	0.848	2.197
Pop11	5	3.634	0.867	0.691	1.346
Pop12	9	6.452	0.855	0.824	1.972
Pop13	3	2.821	0.844	0.609	1.025
Pop14	11	6.972	0.816	0.833	2.103
Pop15	11	7.524	0.797	0.846	2.120
Pop16	11	7.123	0.766	0.831	2.079
Pop17	11	7.730	0.840	0.851	2.156
Mean	10	7.115	0.861	0.824	2.035

**Table 4 genes-13-02383-t004:** AMOVA of the original population.

Source of Variation	Degree of Freedom	Sum of Square	Mean of Square	Estimated Variance	Percentage of Variation (%)	Fst	*p*
Among Populations	16	771.821	48.239	1.338	4%	0.042	<0.001
Within Populations	215	6500.756	30.236	30.236	96%	—	—
Total	231	7272.578	31.576	31.574	100%	—	—

**Table 5 genes-13-02383-t005:** Percentage of trait differences between the core and original collection under different sampling strategies.

Sampling Ratio	10%	15%		20%		25%	30%	35%	40%	
Core Collection	B2C3D2	B2C2D3	B2C3D3	B2C3D3	B1C1D2	B2C3D3	B1C2D2	B1C2D2	B1C2D2	B2C1D2
MD (%)	0.0000	0.0000	0.0000	0.0000	0.0000	0.0000	0.0000	0.0000	0.0000	0.0000
CR (%)	93.4255	100.000	100.000	100.000	91.4811	100.000	94.3496	94.5046	94.5046	98.7887
VD (%)	66.6667	60.0000	66.6667	53.3333	66.6667	46.6667	53.3333	46.6667	33.3333	40.0000
VR (%)	141.037	136.418	136.026	128.870	125.758	122.641	120.205	118.457	115.168	113.5148
Numbers	21	32		42		53	64	73	84	

The meanings of B1, B2, C1, C2, C3, D1, D2 and D3 in the table are shown in the Figure 6.

**Table 6 genes-13-02383-t006:** Genetic parameters of the original and core collections.

Software	Core Collection	Sample Number No.	Allele Number Na	Effective Number of Allele Ne	Observed Heterozygosity Ho	Expected Heterozygosity He	Shannon Information Index I	Polymorphism Information Content PIC
	Original collection	232	26.188	11.565	0.863	0.897	2.660	0.8890
Core Finder	Mc1	158	26.156	11.826	0.863	0.899	2.687	0.8916
Core Hunter	H-10	23	15.000 **	9.974	0.777 **	0.887	2.436	0.8770
	H-15	35	17.531 **	10.684	0.784 **	0.896	2.537	0.8875
	H-20	46	18.719 **	11.046	0.792 **	0.898	2.572	0.8901
	H-25	58	20.031 **	11.234	0.798 **	0.899	2.603	0.8912
	H-30	70	21.375 **	11.439	0.811 *	0.900	2.629	0.8924
	H-35	81	21.938 **	11.379	0.820 *	0.900	2.631	0.8920
	H-40	93	22.719 *	11.411	0.826	0.900	2.641	0.8920
	H-45	104	22.875 *	11.481	0.833	0.900	2.644	0.8921
	H-50	116	23.188 *	11.479	0.837	0.899	2.644	0.8912
	H-55(MC2)	128	23.594	11.483	0.840	0.899	2.645	0.8909
	H-60	139	24.219	11.572	0.839	0.899	2.656	0.8916

* *p* ≤ 0.05 or ** *p* ≤ 0.01 for the difference between a core subset and the original collection in independent *t*-test.

**Table 7 genes-13-02383-t007:** Results of the *t*-test for the diversity parameters of the core and original collections.

Germplasm	N	Na	Ne	Ho	He	I	PIC
Original Collection	232	26.188	11.565	0.863	0.897	2.660	0.8890
Mc1	158 (68.1%)	26.156 (99.9%)	11.826 (102%)	0.863 (100%)	0.899 (100%)	2.687 (101%)	0.8916 (100%)
Mc2	128 (55.2%)	23.594 (90.1%)	11.483 (99.3%)	0.840 (97.3%)	0.899 (100%)	2.645 (99.4%)	0.8909 (100%)
*t* _1_		0.98	0.83	0.97	0.84	0.75	0.84
*t* _2_		0.08	0.94	0.85	0.21	0.87	0.87

N is the total number of samples; *t*_1_ represents the *t*-test value of core collection Mc1 and original collection, and *t*_2_ represents the *t*-test value of core collection Mc2 and original collection.

## Data Availability

All data necessary for confirming the conclusions of the article are present within the article, figures and tables, and within Appendix A.

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
