# Peer review of "Genetic Diversity and Population Structure Analysis of Castanopsis hystrix and Construction of a Core Collection Using Phenotypic Traits and Molecular Markers"

_genes, 2022, doi:10.3390/genes13122383_

Round 1
Reviewer 1 Report
I reviewed the manuscript “Genetic Diversity, Population Structure Analysis of Castanopsis hystrix and Construction of a Core Collection Using Phenotypic Traits and Molecular Markers" submitted to the journal Genes. The manuscript presents some new and interesting data on population structure and core collection of Castanopsis hystrix in China.
I recommend the publication of the paper after some modifications and improvements.
My comments are:
Lines 19 -22: The results part in this abstract is short, so I recommend you to add other data because the results section is the main section in the abstract.
Line 20: Please correct “srrong”
Lines 23 and 24: These sentences are related to the methodology, therefore, transfer them before line 19 or remove them. Line 35: information NOT Information
Line 41: I think we cannot say “Recently” because these references from more than 30 years ago.
Line 47: Please correct “Brown and Frankel et al.[7-9]” to be Brown (7 and 8)and Frankel and Brown (9).
Line 63: What is the meaning of “a wide range of materials?”
Lines 71-73: Add reference, please.
Lines 74 and 75: Simple sequence repeats (SSR) was mentioned before that in line 72, therefore, here mention only the abbreviation.
Lines 74-79: A long sentence, so split it.
Line 81: Simple sequence repeats (SSR) was mentioned before in line 72, therefore, here mention only the abbreviation.
Line 96: What are these phenotypic traits? How did you take these samples? What criteria were taken into account during selecting samples?
Line 97: What is the meaning of the letters in the first row in Table S1. Please explain that in the footnote of the Table.
Lines 101 and 102: More details are requested.
Line 142: diversity NOT giversity.
Line 55: average NOT aver-age.
Line 157: What is the meaning of “red cone populations” in this paragraph?
Line 159: Why did you mention that “the lowest genetic diversity was SSR10”?? Although there are other SSR markers with the same number of alleles.
Lines 171 – 173: This sentence should be moved to the discussion section. Line 174: populations NOT population.
Lines 180 and 181: The authors mentioned “The three subgroups were designated as Q1, Q2, 180 and Q3 (indicated in blue, red, and green, respectively, in Figure. 2)” Is this order correct? I mean Q number and color, please check.
Line 194: Figure 3a or Figure 3 and where is the Figure 3b?
Line 202: In Table S4, there are some Chinese letters, please remove.
Line 237: Figure 5 is not completely clear to the readers, please add high quality figure. Also, the authors mentioned the yellow color in the key of the figure, but I cannot see any indication for this color in most parts of this figure, please check.
Lines 255-259: A long sentence, please split it.
Lines 265-269: A long sentence, please split it.
Line 275: What is the indication of the superscript letters, No.a , Nab, ….
Line 304: What is the difference between the “scientists and breeders”? Are breeders not scientists?
Lines 310 and 311: Why SSR markers, was found to be the optimal approach? Add reference.
Lines 295 – 335: The authors mentioned the importance of molecular markers, genetic diversity …etc, but they did not mention what are the reasons of genetic diversity among and within the 232 accessions. So this part of discussion is very poor.
Line 343: Please, correct “relation-ships”
Lines 364 – 366: WHY? Please add reasons for these information.
Line 377: We NOT we
Line 385: However NOT however
Lines 386 and 387: These information were repeated many time in different sections of the manuscript. Please check that entire the paper and minimize the duplication of information.
Lines 391 – 399: The authors mentioned results not discussion for the results. Actually, the discussion section is very poor, therefore, I recommend authors to add real reasons to clearly explain the obtained results.

Reviewer 2 Report
The authors have done important work and obtained interesting results. It is important to note the comprehensive approach that they used in their work. However, the authors need to pay more attention to the design of the article and the presentation of the results.
Introduction:
Lines 70-73
«In the past, molecular markers, i.e., random amplified polymorphic DNA (RAPD), inter-simple sequence repeat (ISSR) and simple sequence repeats (SSR) have been applied to assess the genetic diversity of C. hystrix resources.» - it is necessary to add references
Material and methods:
It’s necessary to describe in more details how accessions of C. hystrix were grown at experimental site. Do they planted simultaneously or do they have different age? How was they collected and planted to the experimental site? What authors mean by “clones” and how do they obtained this clones? If these are vegetative clones of the same plant, then this is good for assessing the phenotype, but then they should not be considered as independent samples when assessing genetic diversity. Otherwise probably will be better don’t use “clones”. All this details will affect the interpretation of results.
It would be useful to add the map that will reflect areal of C. hystrix and collection sites to visualize how do sampling reflects the native range of distribution of C. hystrix.
Lines 98-102
«In this study, 32 pairs of primers with high polymorphism and good reproducibility were selected from screening a pool of (123 pair) C. hystrix SSR primers in the early stage for analysis(Table S2). For the specific information of primers, genomic DNA extraction methods and SSR amplification procedures, please refer to the previous study of Ms.Yang (Table S3)»
- If the results of previous study of Ms.Yang were published somewhere it is necessary to provide the citation of the published manuscript. Otherwise it should be described in details in the materials and methods – how were this SSR markers obtained, tested and analyzed.
Results:
All the accessions were ordered against the first, second and three factor, showing no clear evidence of a correlation with their geographical origin. – It will be better to reformulate the sentence
Figure 3. – I suggest to remove sample numbers from the picture
Figure 5. – too small, I suggest to move the figure to supplementary but make it bigger.
Table 5.- It is not clear why only several variants but not all of them were included into the table
3.3. Construction of core collection – It is better not to use only abbreviation in the text, It make it hard to understand. For example “had a MD of < 20%, and CR > 80%,” will be better to replace by “had a mean difference (MD)of < 20%, and coincidence rate of range (CR) > 80%”
Line 286
Principal Coordinates Analysis (PCoA) was used… - It is necessary to clarify what data (SSR or phenotypic) were used for PCoA
Figure 6. – It is necessary to indicate which one of core collections was reflected on the figure and was this analysis based on SSR or phenotypic data
Lines 287-291
The results showed that the core collection could well reflect the distribution of the original collection in the principal coordinate, indicating that the constructed core collection is a good representation of the original collection.
- Can’t agree. The picture shows that there is a bias in the core-collection, and the upper right part and the lower right corner are underrepresented. Therefore, it is necessary to discuss this issue in the manuscript.
Discussion
Lines 338 - 339
All three population structure approaches, i.e., STRUCTURE, UPGMA, and principal coordinate analysis, showed a less correlation between population structure and geographic ecotypes.
- In fact there are no data presented about which population belongs to each of STRUCTURE- and UPGMA- groups. And does different samples within populations belongs to different groups. And even about correspondence between STRUCTURE- and UPGMA- groups. I suggest try to add this information to the text and to the figures and make table that will summarize data about all accessions – population, STRUCTURE- and UPGMA- groups and core-collections.
Supplementary Tables
Table S1 List of 232 clones of C. Hystrix – It is not just a list of samples, it is phenotypic data, I guess
Table S3 32 pairs of SSR primer information of C. hystrix 17 populations – It’s necessary to add description of table content
Table S4 Three groups obtained from the STRUCTURE analysis result - It’s necessary to add description of table content and all text should be in English
Round 2
Reviewer 1 Report
Dear authors
Thank you for your response.